# Long-Term Application of Pig Manure to Ameliorate Soil Acidity in Red Upland

Peisang Luo [1], Zedong Long [1], Mei Sun [1], Qiufen Feng [1], Xibai Zeng [2], Hua Wang [3], Zunchang Luo [1] and Geng Sun [1,3,*]

[1]  Hunan Soil and Fertilizer Institute, Hunan Academy of Agricultural Sciences, Changsha 410125, China; luops@grgtest.com (P.L.); longzd313@hunaas.cn (Z.L.)
[2]  Institute of Environment and Sustainable Development in Agriculture, Chinese Academy of Agricultural Sciences, Beijing 100081, China
[3]  College of Environment and Ecology, Hunan Agricultural University, Changsha 410128, China
*   Correspondence: sungeng@hunaas.cn

**Abstract:** While the application of manure to improve soil quality has attracted attention, the effect of pig manure application rates on soil acidity remains poorly understood. In this study, we analyzed the changes and correlations between soil acidity, pH buffer capacity (pHBC), soil chemical properties, and crop yields after 8 years of pig manure application at different rates (i.e., 0, 7.5, 15, 30, and 45 Mg ha$^{-1}$) in a red upland soil (Ultisol). With an increase in the application rates, the crop yields were 0.77–8.85 times higher; the pH was enhanced by 0.4–0.8 units; and the soil organic matter (SOM), pHBC, iron activation (Feo), exchangeable calcium (ExCa), and exchangeable magnesium (ExMg) contents increased by up to 74.1%, 92.7%, 36.7%, 62.2%, and 48.7%, respectively, whereas that of total exchangeable acid (ExAcid) decreased by 17.2–52.9%. The crop yields were positively related to the soil pH but negatively correlated with ExAcid. Redundancy analysis revealed ExAcid and pHBC were more sensitive than pH was to the other chemical indicators. ExAcid was negatively correlated with SOM and ExCa; pHBC was positively correlated with ExMg, TN, SOM, and Feo. In conclusion, the crop yield could be improved by adjusting the soil acidity characteristics, and the application of pig manure reduced the soil acidity, with an optimal application rate of 15 Mg·ha$^{-1}$.

**Keywords:** pig manure; red upland; soil acidity; soil fertility





## 1. Introduction

Red soil (Ultisol) is the most important form of arable land in China, accounting for one-fifth of the total tropical and subtropical areas in South China and producing 44.5% of the total grain yield in the country [1,2]. The acidification of red soil regions has recently intensified due to acid deposition and the overuse of chemical nitrogen fertilizer, which increases aluminum toxicity and decreases nutrient availability, thus threatening soil biodiversity and agricultural production [3–7]. In the last four decades (1980–2020), 145 Tg C (1.1 Mg ha$^{-1}$) of soil inorganic carbon stocks from Chinese croplands has been depleted by N-induced soil acidification through CaCO$_3$ dissolution [8]. Recent research has shown that acidification largely suppresses the natural capacity of the soil microbiome to fight pathogenic fusarium infections, which could play a critical role for plant health [9,10]. Corn planting in red upland areas has increased in recent years in places including Hunan and Jiangxi Provinces, China. However, after the corn harvest, spinach and other winter crops could be planted for an additional income, especially on the outskirts of the city. Therefore, more information is needed on soil acidification and yield increase in maize–vegetable crop rotation.

As the hilly red soil region is an important area for swine husbandry in subtropical and tropical southern China, only in Hunan and Jiangxi Provinces, the total amount of pig manure is more than $0.82 \times 10^{11}$ kg (fresh weight) [11], which provides plentiful available

resources for arable land fertilization and amelioration. The existing research has largely focused on ameliorating soil acidity, decreasing acidic deposition, and inhibiting proton production from synthetic nitrogen fertilizers [12–15]. Organic fertilizer application is an effective agricultural management measure for preventing soil acidification and increasing soil fertility. The long-term application of manure has been shown to prevent and control acidification by increasing the organic matter content and reducing the accumulation of ammonium and nitrate in the soil [16,17]. The application of organic fertilizers in acidic soil can enhance the soil pH, improve soil fertility, promote nutrient absorption by crops, and thus, increase crop yields [18,19]. The long-term application of additional pig manure can prevent red soil acidification and improve phosphorus availability [20]. The soil pH buffer capacity (pHBC) is an important indicator of soil acidification resistance. Manure application increases the soil pHBC and soil resistance to acidification due to the protonation of organic anions from weakly acidic functional groups in soil organic matter and the resulting formation of neutral molecules [6].

In summary, manure application can restrain soil acidification by returning base cations to soils, the ammonification of labile organic nitrogen, the decarboxylation of organic anions, improvements in the pHBC, and the complexation of aluminum with organic matter [6,21,22]. However, there are few studies on the effect of different pig manure application rates on red soil fertility, acidity, and crop yields. To address this knowledge gap, we collected samples of red upland soil in a long-term field experiment and conducted an analysis of the effects of different pig manure application rates on soil fertility, acidity, and crop yields. Our results could provide theoretical guidance for agricultural production in the red soil region.

## 2. Materials and Methods

### 2.1. Experimental Design

The field experiment began in 2013 in Matang Town, Yueyang City, Hunan Province, China (112°44′14″ E, 28°57′11″ N), and was affiliated with the Agricultural Environmental Monitoring Experimental Station of the Ministry of Agriculture and Rural Affairs. The upland soil contained Quaternary red clay (Ultisol). The initial main chemical properties of the soil (0–20 cm) were as follows: pH (soil–water ratio 1:2.5, $w/v$): 5.9; soil organic carbon (SOC): 5.3 g kg$^{-1}$; total nitrogen (TN): 0.52 g kg$^{-1}$; total phosphorus (TP): 0.31 g kg$^{-1}$; total potassium (TK): 12.6 g kg$^{-1}$; alkali-hydrolyzable nitrogen (AN): 42.0 mg kg$^{-1}$; available phosphorus (AP): 5.94 mg kg$^{-1}$; and available potassium (AK): 146.5 mg kg$^{-1}$. A corn–spinach system was employed for the study. Five treatments were tested: pig manure application rates of 0 Mg ha$^{-1}$, 7.5 Mg ha$^{-1}$, 15 Mg ha$^{-1}$, 30 Mg ha$^{-1}$, and 45 Mg ha$^{-1}$. No chemical fertilizers were applied in the experiment. Pig manure was sourced from intensive farms around the experiment site and was applied to the soil in October each year after complete decomposition. The application rate was calculated on a fresh weight basis. Corn straw was removed from the plots. The soil was tilled by hand at a depth of 10 cm. The initial properties of the pig manure (fresh basis) were as follows: pH: 9.66; water content: 47.2%; organic matter: 720.0 g kg$^{-1}$; nitrogen: 14.2 g kg$^{-1}$; phosphorus: 44.1 g kg$^{-1}$; potassium: 21.6 g kg$^{-1}$; calcium: 22.3 g kg$^{-1}$; and magnesium: 9.4 g kg$^{-1}$. The total nutrients and cations applied under different treatments are listed in Table 1. A complete randomized group arrangement was utilized, with three replications of each treatment applied to 21 m$^2$ plots.

The spinach and corn varieties were JieSheng and ChuangYu 118, respectively. The spinach sowing took place in late November in lines spaced 10 cm and rows spaced 15 cm apart. The spinach was harvested in early March of the following year. The corn was sown in April in lines spaced 30 cm and rows spaced 60 cm apart, and harvested in August. Each plot was manually weeded. Pesticides were applied during the growth period as needed. All aboveground crop biomass was removed from the plot following crop harvest.

**Table 1.** Nutrients and base cations applied under different pig manure application rates.

| Application Rates (Mg ha$^{-1}$) | Input (Mg ha$^{-1}$) | | | | | |
|---|---|---|---|---|---|---|
| | **OM** | **N** | **P** | **K** | **Ca** | **Mg** |
| 0 | 0.0 | 0.00 | 0.00 | 0.00 | 0.00 | 0.00 |
| 7.5 | 2.9 | 0.06 | 0.17 | 0.09 | 0.09 | 0.04 |
| 15 | 5.7 | 0.11 | 0.35 | 0.17 | 0.18 | 0.07 |
| 30 | 11.4 | 0.22 | 0.70 | 0.34 | 0.35 | 0.15 |
| 45 | 17.1 | 0.34 | 1.05 | 0.51 | 0.53 | 0.22 |

Note: OM, organic matter; N, nitrogen; P, phosphorus pentoxide; K, potassium oxide; Ca, calcium; Mg, magnesium.

### 2.2. Crop Yield Measured

In 2020 and 2021, during the spinach-picking period, all plants were harvested in each plot. Then, the flesh weight was calculated. At corn maturity, a subsample of plants was harvested from each plot. Then the grains from every plot were dried and weighed to calculate the standard grain yield. The water contents in grains of corn were lower than 13.5%.

### 2.3. Soil Sampling and Soil Physicochemical Analysis

Soil samples were collected in October 2021 before pig manure was applied. Topsoil (0–20 cm) was collected in each plot using the 5-point mixing sampling method. The soil was air-dried after removing plant roots and small rocks.

Soil TN, AN, TP, AP, TK, and AK were measured using conventional methods. Soil AP was extracted using sodium bicarbonate and determined by molybdenum antimony resistance spectrophotometry. AK was extracted using ammonium acetate and measured by flame photometry [23]. SOC was quantified using an organic carbon analyzer (Shimadzu, Kyoto, Japan). Soil pH was measured using a pH meter (Mettler Toledo, Zurich, Switzerland) at a soil/water ratio of 1:2.5. Soil acid buffer capacity (pHBC) was determined by an acid–base titration method [24]. Exchangeable hydrogen (ExH) and exchangeable aluminum (ExAl) were determined by NaOH-neutralization titration. Exchangeable acidity (ExAcid) was quantified by ExH and ExAl. Exchangeable potassium (ExK), exchangeable sodium (ExNa), exchangeable calcium (ExCa), and exchangeable magnesium (ExMg) were extracted with ammonium acetate and titrated complexometrically and then determined through inductively coupled plasma atomic emission spectroscopy (Thermo Jarrell Ash Ltd., Boston, MA, USA). Free iron oxide (Fed) was extracted with a sodium dithionite–sodium citrate–sodium bicarbonate mixed solution. Amorphous iron oxide (Feo) was extracted with an oxalic acid–ammonium oxalate mixed solution. The activation degree of iron oxide was calculated using the ratio of Feo/Fed [25].

### 2.4. Statistical Data Analysis

SPSS 23 (IBM Corp., Armonk, NY, USA) software was used to analyze differences between treatments, with a significance threshold of $p < 0.05$. Pearson correlation analysis was used to analyze the correlation between soil acidity characteristics and crop yields using Origin 2018 (OriginLab Corporation, Northampton, MA, USA). The "vegan" package of the R statistical software (version 4.2.3) was used for redundancy analysis (RDA). Results in the graphs and tables in this paper are expressed as mean values $\pm$ SD.

## 3. Results

### 3.1. Crop Yields

Overall, both spinach and corn yields increased with the increase in the pig manure application rate (Table 2). In 2020, under rates of 7.5, 15, 30, and 45 Mg ha$^{-1}$, spinach yields increased by 79.2%, 205.3%, 272.9%, and 313.5%, respectively, and corn grain yields increased by 77.3%, 238.7%, 310.1%, and 381.5%, respectively, relative to the rate of 0 Mg ha$^{-1}$. In 2021, under rates of 7.5, 15, 30, and 45 Mg ha$^{-1}$, spinach yields were 0.98, 2.43, 3.13, and 3.88 times higher, respectively, and corn grain yields were 2.65, 5.33, 6.30, and 8.85 times

higher, respectively, in comparison to no pig manure inputs. There was a significant difference between all treatments, demonstrating that the application of pig manure enhanced crop yields.

**Table 2.** Spinach and corn yield under different pig manure application rates.

| Application Rates (Mg ha$^{-1}$) | 2020 | | 2021 | |
|---|---|---|---|---|
| | Spinach | Corn | Spinach | Corn |
| 0 | 2.07 ± 0.11 [e] | 1.19 ± 0.22 [e] | 1.65 ± 0.09 [e] | 0.46 ± 0.04 [d] |
| 7.5 | 3.71 ± 0.30 [d] | 2.11 ± 0.14 [d] | 3.26 ± 0.15 [d] | 1.22 ± 0.10 [c] |
| 15 | 6.32 ± 0.15 [c] | 4.03 ± 0.18 [c] | 5.66 ± 0.28 [c] | 2.91 ± 0.56 [b] |
| 30 | 7.72 ± 0.14 [b] | 4.88 ± 0.07 [b] | 6.81 ± 0.34 [b] | 3.36 ± 0.33 [b] |
| 45 | 8.56 ± 0.17 [a] | 5.73 ± 0.90 [a] | 8.05 ± 0.25 [a] | 4.53 ± 0.83 [a] |

Note: Different letters indicate significant differences (LSD test, $p < 0.05$). The values represent the mean ± SD ($n = 3$).

### 3.2. SOM Content

SOM content trended upwards with increases in the application rate of pig manure. Under application rates of 7.5, 15, 30, and 45 Mg ha$^{-1}$, SOM contents were 14.3, 15.3, 18.7, and 19.5 g kg$^{-1}$, respectively, representing changes of 27.7%, 36.6%, 67.0%, and 74.1%, relative to a rate of 0 Mg ha$^{-1}$ (Figure 1). SOM contents were raised significantly at the rates of 7.5 and 15 Mg ha$^{-1}$. Meanwhile, SOC in soil treated with 30 and 45 Mg ha$^{-1}$ was significantly higher than that of 7.5 and 15 Mg ha$^{-1}$.

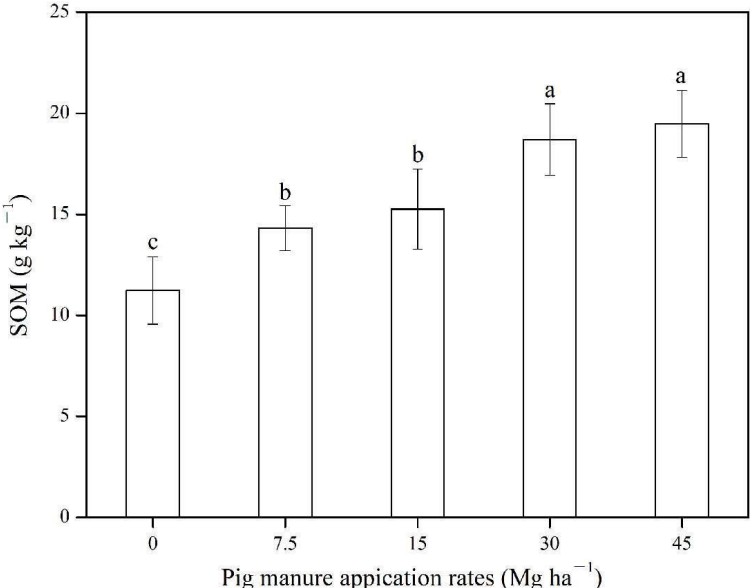

**Figure 1.** SOM contents under different pig manure application rates. Different letters indicate significant differences (LSD test, $p < 0.05$).

### 3.3. Soil Nutrients

The contents of the other soil parameters exhibited the same tendency: AP was particularly sensitive, increasing to 4.5–16.9 times the level under no pig manure inputs. Under the rate of 15 Mg ha$^{-1}$, TN, AN, and AK were significantly enhanced by 27.0%, 37.0%, and 57.3%, respectively. Under the rate of 45 Mg ha$^{-1}$, TN, TP, AN, and AK were 1.40 g kg$^{-1}$, 1.08 g kg$^{-1}$, 97.0 mg kg$^{-1}$, and 327.0 mg kg$^{-1}$, respectively, representing striking increases of 57.3%, 116.8%, 71.2%, and 87.2%. Changes in TK among the treatments were not significant (Table 3).

**Table 3.** Soil nutrients under different pig manure application rates.

| Application Rates (Mg ha$^{-1}$) | TN | TP | TK | AN | AP | AK |
|---|---|---|---|---|---|---|
| | (g kg$^{-1}$) | | | (mg kg$^{-1}$) | | |
| 0 | 0.89 ± 0.05 [d] | 0.50 ± 0.11 [b] | 12.9 ± 0.6 [a] | 56.7 ± 4.67 [b] | 8.13 ± 4.98 [e] | 174.7 ± 13.8 [d] |
| 7.5 | 1.09 ± 0.05 [c] | 0.47 ± 0.19 [b] | 13.1 ± 1.0 [a] | 62.7 ± 7.69 [b] | 36.2 ± 9.82 [d] | 213.3 ± 24.6 [cd] |
| 15 | 1.13 ± 0.10 [bc] | 0.67 ± 0.28 [ab] | 12.5 ± 0.8 [a] | 87.7 ± 2.40 [a] | 62.4 ± 18.4 [c] | 235.3 ± 24.8 [bc] |
| 30 | 1.32 ± 0.18 [ab] | 0.70 ± 0.45 [ab] | 12.9 ± 1.0 [a] | 86.0 ± 8.72 [a] | 91.9 ± 14.9 [b] | 268.0 ± 12.5 [b] |
| 45 | 1.40 ± 0.10 [a] | 1.08 ± 0.21 [a] | 13.1 ± 0.4 [a] | 97.0 ± 7.04 [a] | 137.5 ± 13.0 [a] | 327.0 ± 36.4 [a] |

Note: TN, total nitrogen; TP, total phosphorus; TK, total potassium; AN, alkali-hydrolyzable nitrogen; AP, available phosphorus; AK, available potassium. Different letters indicate significant differences (LSD test, $p < 0.05$).

### 3.4. Soil Exchangeable Acidity

Soil exchangeable acidity (ExH and ExAl) tended to decrease with an increase in pig manure application. Compared with control treatment, ExH content under the rates of 7.5, 15, 30, and 45 Mg ha$^{-1}$ declined notably (18.1%, 28.5%, 41.7%, and 49.3%, respectively), but there was no significant difference between the treatments of 7.5 and 15 Mg ha$^{-1}$ or 30 and 45 Mg ha$^{-1}$. In addition, ExH content under the rates of 30 and 45 Mg ha$^{-1}$ was significantly higher than that of 7.5 and 15 Mg ha$^{-1}$. ExAl content reduced dramatically under 30 and 45 Mg ha$^{-1}$ (43.5% and 52.9%, respectively), but was not significantly decreased under 7.5 and 15 Mg ha$^{-1}$ (Figure 2).

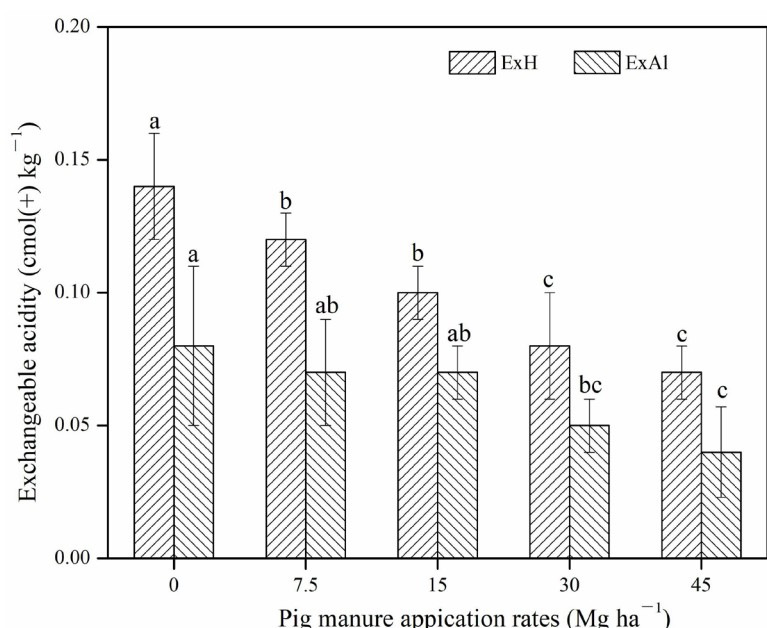

**Figure 2.** Soil exchangeable acid under different manure application rates. Note: ExH, exchangeable hydrogen; ExAl, exchangeable aluminum. Different letters indicate significant differences (LSD test, $p < 0.05$).

### 3.5. Soil Exchangeable Base Cations

The content of soil exchangeable base cations (i.e., ExK, ExNa, ExCa, and ExMg) generally increased with the amount of pig manure applied. Under the rate of 45 Mg ha$^{-1}$, ExK, ExNa, ExCa, and ExMg contents were 0.83, 1.83, 12.2, and 5.6 cmol kg$^{-1}$, respectively, representing significant increases of 59.6%, 61.8%, 62.2%, and 48.7%, respectively, compared with no pig manure inputs. The contents of ExNa, ExCa, and ExK under a rate of 30 Mg ha$^{-1}$ also increased significantly (45.5%, 32.1%, and 54.2%, respectively). There was no significant difference between 0, 7.5, and 15 Mg ha$^{-1}$ (Figure 3).

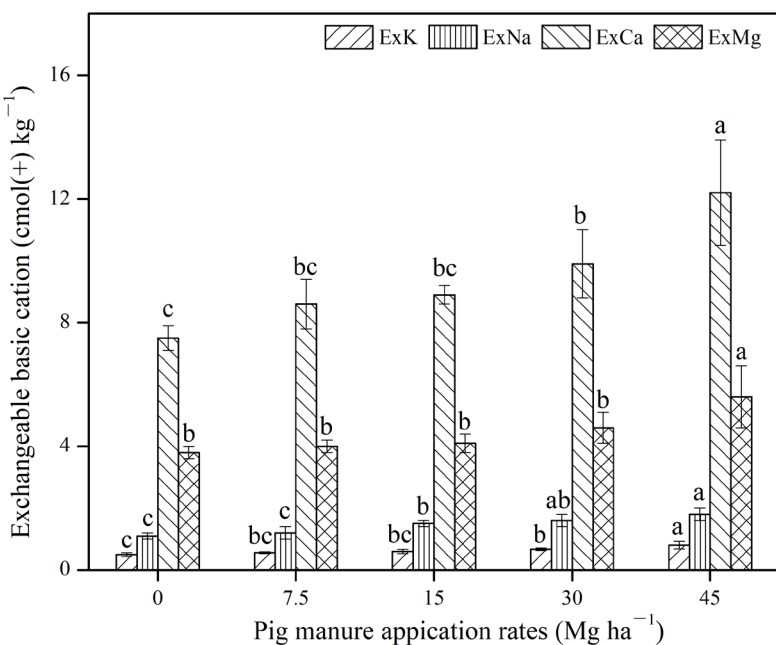

**Figure 3.** Soil exchangeable basic cations under different manure application rates. Note: ExK, exchangeable potassium; ExNa, exchangeable sodium; ExCa, exchangeable calcium, ExMg, exchangeable magnesium. Different letters indicate significant differences (LSD test, $p < 0.05$).

### 3.6. Form and Activation of Iron Oxide

No significant difference was observed in Fed content among all treatments (Figure 4). Feo content under the rates of 7.5, 15, 30, and 45 Mg ha$^{-1}$ was significantly higher than that of 0 Mg ha$^{-1}$ (8.8%, 11.8%, 17.6%, and 26.5%, respectively), and there was no difference between 30 and 45 Mg ha$^{-1}$. In our study, the activation degree of iron increased with increased pig manure application. The ratio of Feo/Fed under the rates of 7.5, 15, 30, and 45 Mg ha$^{-1}$ increased significantly (13.3%, 10.0%, 26.7%, and 36.7%, respectively). There was no significant difference between treatments of 7.5 and 15 Mg ha$^{-1}$ or 30 and 45 Mg ha$^{-1}$, but the ratio of Feo/Fed for soil treated with 30 and 45 Mg ha$^{-1}$ was significantly higher than that for soil treated with 7.5 and 15 Mg ha$^{-1}$ (Figure 4).

### 3.7. Soil pH, Total Exchangeable Acidity, and pHBC

Soil pH tended to increase with an increase in pig manure application (Figure 5). The pH values under the rates of 30 and 45 Mg ha$^{-1}$ were 6.6 ± 0.1 and 6.7 ± 0.1, respectively, which were significantly higher than those for the 0 Mg·ha$^{-1}$ (5.9 ± 0.2) and 7.5 Mg ha$^{-1}$ (6.3 ± 0.3) treatments. Soil ExAcid decreased with an increase in pig manure application. Compared with no pig manure inputs, ExAcid under the rates of 15, 30, and 45 Mg ha$^{-1}$ decreased significantly by 25.8%, 41.3%, and 52.9%, respectively, but there was no significant difference between 30 and 45 Mg ha$^{-1}$ treatments. All four manure treatments had significantly higher soil pHBC than no pig manure inputs, as soil pHBC increased by 42.6%, 55.3%, 63.2%, and 92.7% under the rates of 7.5, 15, 30, and 45 Mg ha$^{-1}$, respectively. There was no significant difference in soil pHBC between 30 and 45 Mg ha$^{-1}$. Linear analysis shows that pH and pHBC increased by 0.017 units and 0.26 cmol(+) kg$^{-1}$, respectively, accompanying an application rate increase of 1 Mg ha$^{-1}$. In summary, the long-term application of pig manure clearly improves the pHBC in red soil.

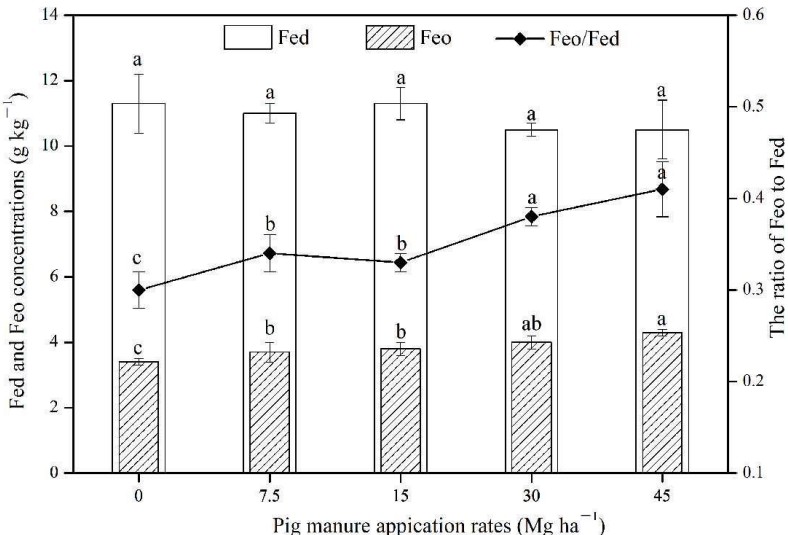

**Figure 4.** Forms and activation of iron oxide under different manure application rates. Note: Fed, free iron oxide; Feo, amorphous iron oxide; Feo/Fed, activation of iron oxide. Different letters indicate significant differences (LSD test, $p < 0.05$).

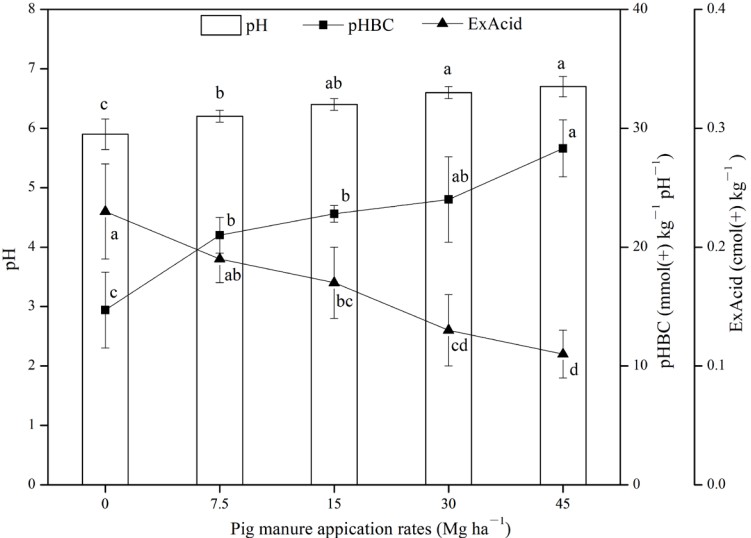

**Figure 5.** Soil pH, pHBC, and ExAcid under different manure application rates. Note: pHBC, pH buffer capacity; ExAcid, total exchangeable acidity. Different letters indicate significant differences (LSD test, $p < 0.05$).

### 3.8. The Effects of Soil Acidity Characteristics on Crop Yields

There were positive correlations between soil acidity characteristics and crop yields (Figure 6). Spinach and corn yields decreased along with an increase in soil ExH and ExAl in both 2020 and 2021. The yields of spinach and corn increased along with an increase in soil pH and pHBC in both 2020 and 2021.

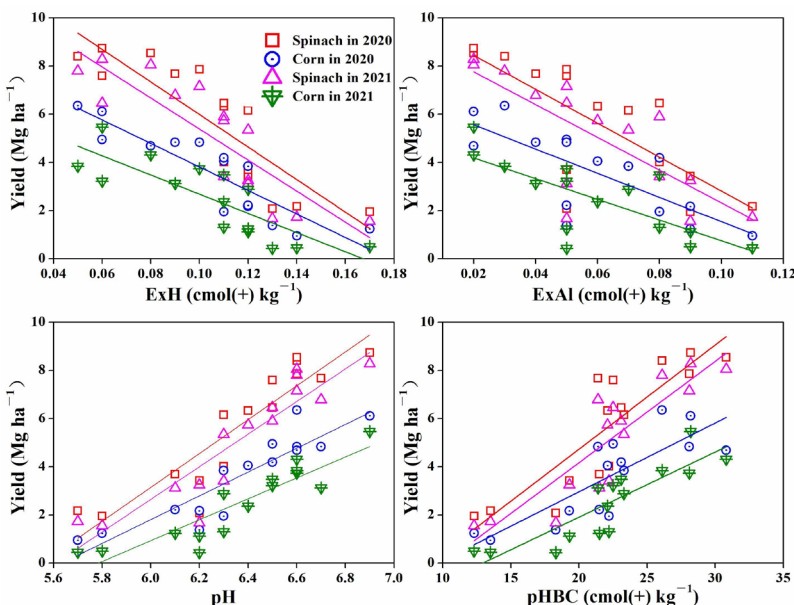

**Figure 6.** The relationships between soil acidity characteristics and crop yields.

The relationship between soil acidity characteristics and crop yields could be fit by linear equations with $R^2 > 0.50$ and $p < 0.01$ (Table 4). The slopes of the linear equations showed that spinach and corn yields decrease (0.67–0.70 and 0.42–0.51 Mg ha$^{-1}$) with an increase in soil ExH (1.0 cmol(+) kg$^{-1}$). Meanwhile, yields of spinach and corn would decrease 0.73–0.75 and 0.45–0.53 Mg ha$^{-1}$ with a soil ExH increased by 1.0 cmol(+) kg$^{-1}$. However, spinach and corn yield increases of 0.68–0.70 and 0.43–0.49 Mg ha$^{-1}$ would accompany a soil pH increase of 0.1 unit. Furthermore, yields of spinach and corn would increase by 0.42–0.43 and 0.27–0.29 Mg ha$^{-1}$ with a soil pHBC increase of 1.0 cmol(+) kg$^{-1}$.

**Table 4.** The linear equations between soil acidity characteristics and crop yields.

| Acidity Characteristics | Crop Yield | Intercept | Slope | $R^2$ | $p$ |
|---|---|---|---|---|---|
| ExH | Spinach in 2020 | 13.16 | −70.43 | 0.71 | <0.01 |
| | Corn in 2020 | 9.01 | −51.06 | 0.74 | <0.01 |
| | Spinach in 2021 | 12.23 | −67.26 | 0.70 | <0.01 |
| | Corn in 2021 | 6.94 | −41.84 | 0.63 | <0.01 |
| ExAl | Spinach in 2020 | 10.19 | −74.93 | 0.57 | <0.01 |
| | Corn in 2020 | 6.79 | −53.20 | 0.57 | <0.01 |
| | Spinach in 2021 | 9.45 | −72.51 | 0.58 | <0.01 |
| | Corn in 2021 | 5.22 | −45.22 | 0.53 | <0.01 |
| pH | Spinach in 2020 | −39.05 | 7.03 | 0.81 | <0.01 |
| | Corn in 2020 | −27.84 | 4.94 | 0.80 | <0.01 |
| | Spinach in 2021 | −38.01 | 6.78 | 0.82 | <0.01 |
| | Corn in 2021 | −25.10 | 4.34 | 0.80 | <0.01 |
| pHBC | Spinach in 2020 | −3.91 | 0.43 | 0.73 | <0.01 |
| | Corn in 2020 | −2.75 | 0.29 | 0.63 | <0.01 |
| | Spinach in 2021 | −4.25 | 0.42 | 0.76 | <0.01 |
| | Corn in 2021 | −3.50 | 0.27 | 0.74 | <0.01 |

### 3.9. Analysis of Influencing Factors of Soil Acidity

Results of the redundancy analysis of soil acidity characteristics and influencing factors are shown in Figure 7. The first and second axes explained 86.7% and 3.9% of the change in acidity characteristics, respectively. The change in soil acidity characteristics was well explained by the chemical indicators. Soil pH was slightly affected by chemical indicators, while ExAcid and pHBC were greatly sensitive to the chemical indicators. The ExAcid was

positively correlated with Fed and negatively correlated with SOM and ExCa. Soil pHBC was positively correlated with ExMg, TN, SOM, and Feo.

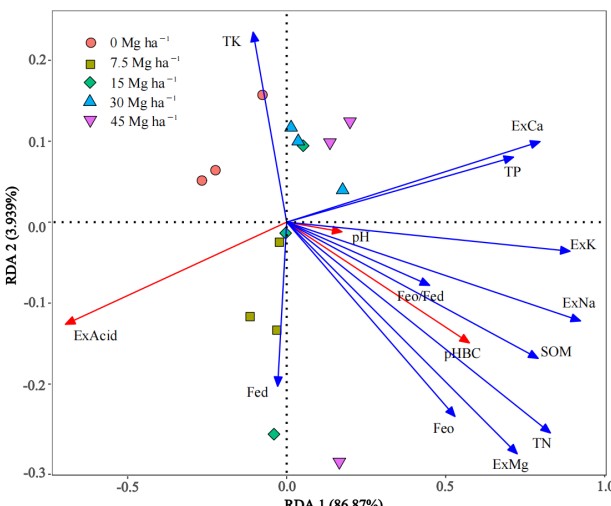

**Figure 7.** Redundancy analysis (RDA) of soil acidity characteristics. Note: TN, total nitrogen; TP, total phosphorus; TK, total potassium; AN, alkali-hydrolyzable nitrogen; AP, available phosphorus; AK, available potassium.

## 4. Discussion

### 4.1. Regulation of Pig Manure on the Acidity of Red Soil

Excessive application of chemical nitrogen in soil can produce hydrogen ions through nitrification, which is a key factor responsible for soil acidification [26]. Our study demonstrates that long-term application of pig manure can effectively regulate the acidity of red soil, resulting in increased soil pH and decreased exchangeable acid (Figure 5). Crop yields were positively related to soil pH but negatively correlated with soil ExH and ExAl (Figure 6). In addition, linear equations indicated that the dropped rates of spinach were higher than those of corn. Our RDA analysis indicated that soil base cations can affect soil pH, exchangeable acid, and acid buffer capacities (Figure 7). Acid deposition and excessive application of chemical nitrogen fertilizer can lead to a large loss of base cations along the soil profile [19,27]. Meanwhile, active hydrogen occupies the cation exchange site, increases the content of soil exchangeable acid, and exacerbates soil acidification [5,28]. The application of manure may prevent soil acidification. Firstly, manure is rich in alkaline substances which can directly neutralize acidogenic ions and effectively replenish exchangeable base cations in the soil [29]. Secondly, a large amount of acidic oxygen-containing functional groups contained in manure could increase the negative charge sites of the soil through carboxyl protonation and enhance soil pHBC, thereby enhancing soil resistance to acidification [6,30,31]. Moreover, manure is effective in adsorbing reactive hydrogen and aluminum, thus reducing total acidity, especially from exchangeable aluminum [16]. Finally, cations such as Ca and Mg combined with the organic anions in manure can increase soil base saturation [32].

### 4.2. Promotion of pHBC in Red Soil by Pig Manure

According to a previous classification [33], the soil buffer system in our study comprised a silicate buffer (pH 8.6–6.2) and calcium carbonate buffer (pH > 5.0). Red soil does not contain calcium carbonate, and silicate buffer dominates soil pHBC [28]. In the process of acidification, the buffering of red soil to exogenous acid is mainly caused by the release of silicate minerals, increasing soil cation exchange capacity and absorption sites on the soil's solid surface [34,35]. In red soil, crop yield could be improved as soil pHBC is increased. Linear equations indicated that the growth rates of spinach were higher than those of corn (Figure 6). The RDA analysis also confirmed that soil pHBC was positively correlated with

soil base cations (Figure 7). The application of manure directly inputs alkaline substances into the soil and increases cation exchange, thus increasing pHBC [6,36]. In addition, manure application can enhance pHBC by increasing soil organic matter content as evaluated by the surface complexation model. Soil surface site concentration, which represented the soil buffer capacity, was positively correlated with organic matter content, and organic matter plays a crucial role in soil pHBC. The RDA analysis results showed soil pHBC to be significantly and positively correlated with organic carbon content (Figure 7). Previous studies have suggested that soil organic matter could buffer exogenous acids by neutralizing and consuming protons through the dissociation of acidic oxygen-containing functional groups (i.e., carboxyl, phenolic hydroxyl, and alcohol hydroxyl). These functional groups exert strong absorption effects on cations, which could reduce the effectiveness of soil cations and increase the content of base cations, thus improving soil pHBC [6,37,38].

### 4.3. Improvement of Red Soil Fertility by Pig Manure

Soil organic matter (SOM) plays a crucial part in sustaining soil fertility and productivity, and the application of organic fertilizers like manure is an effective way to improve soil structure, SOM content, and soil carbon retention, as they increase soil particulate and mineral-associated organic matter conducive to the accumulation of SOM [7,17,39–41]. The application of chemical nitrogen fertilizer, a vital field management strategy, can increase soil productivity in the short term but also can lead to a soil nitrogen surplus [42,43]. The application of swine manure in agroecosystems not only restrains acid but also improves nitrogen cycling efficiency and reduces nitrogen leaching loss [39,44,45]. In addition, the long-term application of manure is beneficial to the absorption of plant-available nitrogen in macroaggregates, and boosting the amount of soluble nitrogen thus increases yields [46]. Under acidic conditions, phosphorus mobility and availability decrease through absorption and fixation, thus decreasing the proportion of phosphorus available to be absorbed and utilized by crops [40,41]. The use of manure fertilizer with a high phosphorus content causes an increase in available phosphorus with the application rate of manure (Table 3). Furthermore, microbial biomass, microbial species richness, microbial activity, and microbial functional diversity have been shown to increase after long-term manure fertilization [45,47,48]. In addition, the application of organic fertilizer can not only increase crop yield but also effectively supplement soil base cations, which is conducive to the sustainable use of soil resources [32]. In tradition, a pig manure application rate of 5–20 Mg ha$^{-1}$ was popular with local farmers. In the present study, crop yields, exchangeable base cations, and soil pH and pHBC increased with an increase in pig manure application rate; soil pH and pHBC were significantly reduced at the rate of more than 7.5 Mg ha$^{-1}$, yet total exchangeable acidity decreased and nutrients (i.e., AN and AK) increased noticeably when the rate was higher than 15 Mg ha$^{-1}$, indicating that 15 Mg ha$^{-1}$ was a recommended application rate. It is well known that improper application of chemical fertilizer can lead to soil acidification. The role of acidity and fertility of red upland under the combination of pig manure and chemical fertilizers warrants further research.

### 5. Conclusions

In the present study, red upland crop (spinach and corn) yields, soil fertility indicators (i.e., soil organic carbon, total nitrogen, total phosphorus, alkali-hydrolyzable nitrogen, and available potassium), soil pH and pH buffer capacity, and exchangeable base cations (i.e., exchangeable calcium, exchangeable magnesium, exchangeable potassium, and exchangeable sodium) increased with the pig manure application rate while exchangeable acid (exchangeable hydrogen and exchangeable aluminum) tended to decrease. Crop yield could be improved by ameliorating soil acidity characteristics. In addition, compared with pH, ExAcid and pHBC were more sensitive to the chemical indicators. Overall, the application of pig manure may reduce soil acidity, increase soil fertility, and raise crop yields. We recommend a manure application rate of 15 Mg ha$^{-1}$ in our studied soil.

**Author Contributions:** Conceptualization, G.S.; methodology, M.S.; software, Z.L. (Zedong Long); formal analysis, M.S. and Q.F.; validation, H.W. and Z.L. (Zunchang Luo); investigation, P.L. and M.S.; resources, X.Z. and H.W.; data curation, M.S. and Q.F.; writing—original draft preparation, P.L.; writing—review and editing, G.S.; visualization, P.L.; supervision, Z.L. (Zunchang Luo) and G.S.; project administration, Z.L. (Zunchang Luo); funding acquisition, X.Z. All authors have read and agreed to the published version of the manuscript.

**Funding:** This research was funded by the Joint Regional Innovation and Development Fund (U19A2048), Hunan Agricultural Science and Technology Innovation Fund Project (2022CX75, 2022CX78), Hunan Province Science and Technology Talent Bootstrap Project (2022TJ–N07).

**Institutional Review Board Statement:** Not applicable.

**Data Availability Statement:** The data that support the findings of this study are available from the corresponding author G.S. (Geng Sun), upon reasonable request.

**Conflicts of Interest:** The authors declare no conflict of interest.

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
