# Peer review of "Long-Term Application of Pig Manure to Ameliorate Soil Acidity in Red Upland"

_agriculture, doi:10.3390/agriculture13091837_

Round 1
Reviewer 1 Report
The manuscript is well-written and discussed nicely. I have only a few observations on the work
Line 41: ‘melioration’. May be amelioration
Line 89-91: Spinach was sown in September and harvested in March of the following year. What is the duration of the Spinach crop in general?
Line 289: Absorption or adsorption?
Please check the units of soil nutrients. May be gkg-1 instead of g.kg-1 or mg.kg-1
Whether the applied doses of pig manure corroborate with the recommended fertilizer doses for corn-spinach? (although this is not directly linked to the results, but may be useful for substantiate the yield of both crops).
Continuous use of pig manure over the years increased the soil pH. Whether continuous use of manure can generate acidity by producing organic acids? If so that information may be included on the discussion part.
Good wishes.
Reviewer 2 Report
The introduction may be improved with the latest state-of-the-art research done.
Use SI units throughout the manuscript e.g. Mg/ha
The results have been described in a precise manner and relevant parameters have been recorded.
How pH of the red soils relate with the crop yield and applied inputs?
I suggest moderate language editing
Reviewer 3 Report
This study investigated the long-term application of different pig manure rates on soil fertility, acidity, and crop yields of corn and spinach in red upland. The results could provide solution for ameliorating soil acidity in red soil region. The topic is interesting. The manuscript is well organized and written. However, there are some minor issues need to addressed before it be accepted for publication. The specific comments are as following:
1. In the Abstract, the results were just some qualitative description, more quantitative descriptions about the results need to be added.
2. Why were TN, AN, TP, AP, TK, and AK these soil properties only measured in 2021? How about the data in 2020?
3. In line 97, what do you mean by “flesh weight”? Is it “fresh weight”? Please check the whole manuscript and correct the mistakes.
4. From line 226-268, “Moreover, soil pH was negatively correlated with ExAcid, pH, and pHBC, and positively correlated with base cations…”, how soil pH could be negatively correlated with pH? As can be seen in Figure 7, soil pH was positively correlated with pHBC but not negatively correlated with it. In addition, I can not get to much information from the redundancy analysis in text, maybe further analysis is needed to added.
The language of the manuscript need to be improved.
